# A benchmarked comparison of software packages for time-lapse image processing of monolayer bacterial population dynamics

Atiyeh Ahmadi,[1] Matthew Courtney,[2] Carolyn Ren,[2] Brian Ingalls[1,3]

**ABSTRACT** Time-lapse microscopy offers a powerful approach for analyzing cellular activity. In particular, this technique is valuable for assessing the behavior of bacterial populations, which can exhibit growth and intercellular interactions in a monolayer. Such time-lapse imaging typically generates large quantities of data, limiting the options for manual investigation. Several image-processing software packages have been developed to facilitate analysis. It can thus be a challenge to identify the software package best suited to a particular research goal. Here, we compare four software packages that support the analysis of 2D time-lapse images of cellular populations: CellProfiler, SuperSegger-Omnipose, DeLTA, and FAST. We compare their performance against benchmarked results on time-lapse observations of *Escherichia coli* populations. Performance varies across the packages, with each of the four outperforming the others in at least one aspect of the analysis. Not surprisingly, the packages that have been in development for longer showed the strongest performance. We found that deep learning-based approaches to object segmentation outperformed traditional approaches, but the opposite was true for frame-to-frame object tracking. We offer these comparisons, together with insight into usability, computational efficiency, and feature availability, as a guide to researchers seeking image-processing solutions.

**IMPORTANCE** Time-lapse microscopy provides a detailed window into the world of bacterial behavior. However, the vast amount of data produced by these techniques is difficult to analyze manually. We have analyzed four software tools designed to process such data and compared their performance, using populations of commonly studied bacterial species as our test subjects. Our findings offer a roadmap to scientists, helping them choose the right tool for their research. This comparison bridges a gap between microbiology and computational analysis, streamlining research efforts.

**KEYWORDS** time-lapse imaging, image processing, image segmentation, tracking, population dynamics

Time-lapse fluorescence microscopy is a powerful technique for studying biological processes at the single-cell level (1–3). This approach is frequently used to investigate the dynamics of bacterial populations (4–9). A common strategy involves imaging in a single focal plane: observations are made of cells growing and interacting in a two-dimensional monolayer on microscope slides or in microfluidic traps (2, 10). These experiments generate large quantities of data, especially when images are collected frequently. Research goals typically involve gathering quantitative data from these images (e.g., cell position, elongation rate, and fluorescence intensity).

Manual extraction of this data is infeasible in most cases, resulting in demand for dedicated image-processing software packages. The community has risen to this challenge: a number of packages have been developed, focusing on a range of cell types, experimental set-ups, and analysis features [reviewed in reference (11)]. Given this range

Address correspondence to Brian Ingalls, bingalls@uwaterloo.ca.

The authors declare no conflict of interest.

See the funding table on p. 17.

of options, identifying the ideal software package for a given research project can be a daunting task. Here, we present a systematic performance comparison of image-processing packages to guide readers in their decision of what tools to adopt.

The recent review of Jeckel and Drescher (11) provides a broad survey of image-processing software packages and techniques for the validation of image analysis results. Here, we focus specifically on populations of rod-shaped bacteria (i.e., bacilli) growing in a monolayer. We thus do not address the tools developed for processing alternate image types, e.g., cells growing in mother machines (12–14), bacterial cells with other morphologies [e.g., spherical (15) and crescent with stalk (16)], or animal cells [e.g., reference (17)].

We present benchmarked results on segmentation and tracking, along with direct comparisons of feature assignment and execution times, based on monoculture time-lapse phase contrast images of unconstrained growth of *Escherichia coli*, *Pseudomonas putida*, and *Xanthomonas campestris* populations and of *E. coli* growth in a microfluidic trap [in which populations can be maintained in monolayer in consistent growth conditions over long periods (18)].

For each data set and software package, we follow recommended-practice pipelines for optimal performance. To assess the robustness of our findings, we also briefly evaluate software performance on images of sub-optimal quality.

In preliminary investigations, we identified four tools that performed well on monolayer bacterial population time-course data: CellProfiler (19, 20), SuperSegger-Omnipose (21), Deep Learning for Time-Lapse Analysis (DeLTA) (22, 23), and Feature-Assisted Segmenter/Tracker (FAST) (24); these are the focus of our analysis. In contrast, we found the performance of several related tools to be not competitive on these data, given that their main performance goal is different: BacStalk (16), CellX (25), Trackmate (15), BactMAP (26), CellShape (27), MM3 (13), MMHelper (14), and MoMA (12). We performed extensive analysis with Oufti (28), which has been recommended for the sort of analysis we perform here (26, 29, 30). However, recognizing that this package's main strength is in fine-grained analysis of cell morphology and intracellular structure, rather than tracking population dynamics, we did not include it in the performance comparison.

To compare these software packages, we manually generated cell counts, reference masks, tracking trajectories, and lineage trees from benchmarking data sets. These serve as ground truth for evaluating the performance of each software package in terms of segmentation and tracking.

In the next section, we introduce the four software packages, after which we compare the foundations of their analysis algorithms. We then present performance comparisons in segmentation, tracking, feature assignment, and execution time.

## Image processing package overviews

CellProfiler 4.2.6 (19, 31) was developed for the analysis of images of eukaryotic cells, but it demonstrates strong performance on images of rod-shaped bacteria. This package offers a range of methods for segmentation as well as a wide selection of image-processing modules, including the deep learning segmentation algorithm Omnipose (32), which we incorporate below in our recommended-practice pipeline. In addition, CellProfiler incorporates several algorithms for tracking, which makes the software applicable in a variety of cell motion and crowding conditions.

SuperSegger-Omnipose 0.4.4 (21) produces a wide range of outputs presented to the user in both a graphical user interface and data files. This package has recently undergone innovation in its segmentation strategy (33), moving from traditional methods to the use of Omnipose (32). This implementation incorporates generalized parameters that have been trained on images of bacilli bacteria in a range of conditions, thus minimizing the need for manual tuning.

DeLTA 2.0.5 (22, 23) performs segmentation and tracking with a U-Net convolutional neural network. It supports parallelization of segmentation and tracking steps on CPU or GPU cores, resulting in accelerated computation.

FAST 2.2 (24, 34) offers in-line parameter manipulation, allowing users to tune tracking performance in real-time. This package generates visual outputs incorporating a range of statistical characterizations.

Table S1 (Appendix) summarizes additional features of the four packages.

## Algorithm overview and comparison

Here, we present an overview of the image-processing algorithms employed by the four packages under consideration. [We refer the reader to reference (11) for a comprehensive treatment of image processing for cellular analysis.] The image-processing pipeline (Fig. 1) begins with segmentation, in which cell objects are distinguished by grouping pixels with similar attributes. This is followed by tracking, which involves a comparison of images across time points to identify persistent objects (i.e., cells undergoing growth and motion) and mother–daughter triads arising via cell division events. The final step is feature assignment, in which geometric and dynamic features of individual cells are determined (e.g., centroid position, length, and elongation rate).

### *Segmentation*

Segmentation of densely packed groups of cells is a challenging task. Because it is the first step in the image-processing pipeline, success in segmentation is critical to overall performance (19). As discussed in reference (11), segmentation algorithms can be classified into two broad categories: traditional or machine learning based.

Traditional segmentation methods include watershed (suitable when objects are touching or overlapping) and thresholding (effective when foreground and background have different average intensities).

CellProfiler provides the option to employ either watershed or thresholding for traditional segmentation and can accept images that have been pre-processed by external segmentation tools such as Ilastik (35). Additionally, this package can incorporate deep learning segmentation tools such as Omnipose (36), described further below.

FAST employs a multi-stage traditional segmentation process. A texture metric (37) is employed to characterize the organization of pixel intensities within an image, after which an adaptive thresholding step (which can account for uneven illumination) detects ridges. Finally, watershed is applied, separating objects that ridge detection failed to distinguish.

Turning to machine learning-based algorithms, U-Net is a widely used deep learning architecture for image segmentation (38). U-Net models are usually trained on input images and corresponding annotated masks (39, 40). Once trained, the models can generate segmentation masks for previously unseen data. Both DeLTA (23) and Omnipose (36) use U-Net-based approaches for segmentation.

DeLTA employs a structure that is faithful to the original U-Net architecture (40) and incorporates a custom loss function and weight maps used for network training. A size filter is applied after segmentation. In contrast, Omnipose [built on the foundations of the earlier tool CellPose (41)] employs a U-net architecture with minor modification by introducing residual blocks, which allow for layer shortcuts and the retention of intricate feature details. It also integrates a style vector, optimizing the network's adaptability to individual image scenarios.

In the performance comparisons below, we distinguish the segmentation performance of SuperSegger-Omnipose from CellProfiler with Omnipose. The implementations differ slightly: CellProfiler provides a minimum size filter that lets users set the smallest allowable pixel count for a mask. Additionally, CellProfiler automatically removes objects in contact with the image border; for analysis in Supersegger, a custom-written script was developed for this task (available at Github and OSF).

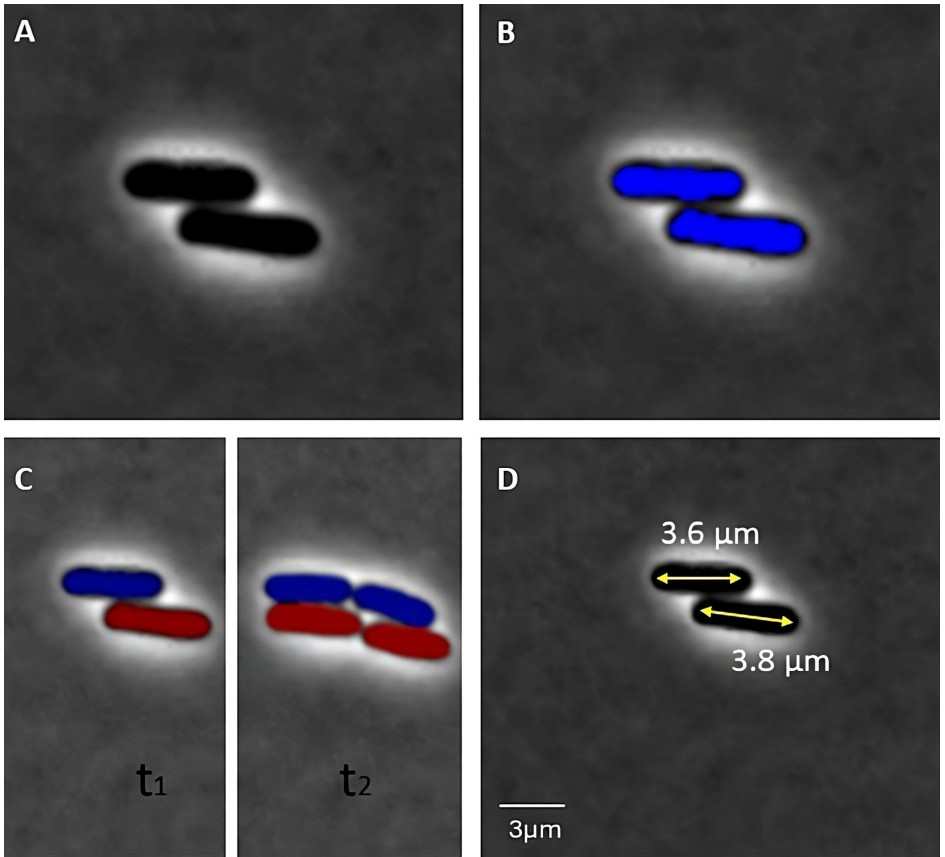

**FIG 1** Image processing pipeline. (A) Raw image. (B) Segmentation: cells are distinguished as individual objects. (C) Tracking: cells are tracked across frames and mother–daughter triads are identified (lineage indicated by color). (D) Feature assignment: geometric features such as length (shown) are determined.

## Tracking

Traditional tracking methods can be classified as follows. In overlap-based tracking, which SuperSegger employs, objects are tracked based on spatial overlap between consecutive frames. In contrast, distance-based tracking relies on the frame-to-frame distance between object perimeters. More generally, feature-based tracking is based on the similarity of features assigned to each object, such as orientation or fluorescence intensity. Finally, in neighborhood-based tracking, objects are matched by coordinating each object's movement with that of its neighbors. CellProfiler offers each of these traditional tracking methods.

DeLTA and FAST use deep learning approaches for tracking: U-Net and unsupervised learning, respectively.

For tracking mother–daughter triads, CellProfiler and SuperSegger use an overlap-based method that records a division event when an object overlaps with two or more objects in the subsequent frame. The procedure for detecting cell division in DeLTA involves training the U-net model; to establish a flag for division events, the algorithm iteratively analyses the complete data set. FAST uses a similar procedure based on unsupervised machine learning.

## Feature assignment

All the packages can extract a wide range of cell object features, such as area (total pixel count within the object's boundary) and phase and/or fluorescence intensity (mean, maximum, and total). In addition, neighborhood-based features provide information

about the cell object's surroundings, such as the number of adjacent cells or the average distance to neighboring cells.

The packages differ somewhat in how they determine the length of bacilli. SuperSegger, CellProfiler, and FAST determine cell length as the length of the major axis of a fitted ellipse. (FAST additionally provides the user the option to determine length as object diameter.) In contrast, DELTA determines cell length as the longer edge of a bounding box (23).

## MATERIALS AND METHODS

### Computational resources

All computational analyses were executed on an Intel(R) Core(TM) i5-7200U CPU with 8 GB of RAM.

### Materials

Bacterial strains used in this study were *E. coli* K-12 (gift from Matthew Scott, University of Waterloo), *Pseudomonas putida* KT2440 (gift from Trevor Charles, University of Waterloo), and *Xanthomonas campestris* (gift from Andrew Doxey, University of Waterloo).

### Experimental methods

#### Culture conditions

Cultures were inoculated from isolated colonies picked from freshly streaked LB agar plates into tubes containing 5 mL LB broth (Fisher Scientific, MA, USA) and incubated overnight at 37°C at 200 rpm, with the exception that *Xanthomonas campestris* was incubated at 27°C. Following overnight growth, each culture was diluted in fresh LB to an $OD_{600}$ of 0.05 and then grown under the same conditions. Samples were taken for microscopy at an $OD_{600}$ of 0.4.

#### Microfluidic chip design and manufacture

A microfluidic device was made using a standard soft lithography multilayer fabrication technique. Briefly, to produce the growth chambers, a layer of SU8 photoresist (2000.5, Microchem, Germany) was spin-coated onto a silicon wafer at 700 rpm. The thickness was measured to be approximately 730 nm using a Dektak 8 stylus profilometer. Next, to produce the flow channel, another layer of SU8 photoresist (2015, Microchem) was spin-coated onto the same wafer with a target thickness of 15 μm. The microfluidic features for each layer were patterned using a photomask and UV light. A notch on each photomask matching the notch of the wafer was used to align the two layers. After baking and developing the photoresist layers, polydimethylsiloxane (PDMS; Krayden, CO, USA) was prepared at a 10:1 base-to-curing agent ratio and then cured on the silicon master mold for 3 hours at 95°C. A 1.5 mm biopsy punch was then used to create inlet and outlet ports. Finally, the PDMS layer was bound to a glass slide using a plasma wand (Electro-Technic Products, IL, USA).

#### Microscopy

Exponentially growing cultures were added to a cover slip-bottomed petri dish (MatTek, MA, USA) and then covered with an agar pad. [To prepare agar pads, 3 g of agar powder (Fisher Scientific, MA, USA) was weighed and added to a mixture of 5 g of LB powder and 200 mL DI water. The dissolved mixture was then autoclaved and cooled to about 50°C. The mixture was poured into petri dishes to a depth of 3–5 mm. After cooling to room temperature, 0.5 cm by 0.5 cm square pads were cut with a scalpel.] The cover slip-bottomed dishes were then placed on the stage of a Zeiss Axio Observer widefield microscope (operated by ZEN2 Blue Edition) equipped with an Axiocam 506

mono camera (Zeiss Axio Observer, Grünauer Fenn, Germany) and then covered with an incubating plate (PECON, Erbach, Germany) to maintain temperature at 37°C (or 27°C for *Xanthomonas campestris*). Phase contrast images were collected—every 1.5 minutes for unconstrained and constrained *E. coli*, every 3 minutes for *P. putida* and *X. campestris*—with a Plan-Apochromat 63/1.4 Oil Ph3 M27 objective (Carl Zeiss Microscopy-LLC, NY, USA). For imaging in microfluidic chambers, exponential phase culture was injected by syringe into the chip inlet. The inlet was then connected to a syringe pump containing fresh LB, and the outlet was connected to a waste vial. The device was then placed on the microscope stage under the incubating chamber. Fresh media were supplied at a flow rate of 2 µL/min.

## Computational methods

### Manual image annotation

All manual analysis was performed by a single team member to ensure consistency.

### Manual counting of cell objects

For each frame, cell counts were manually determined. Cells touching the borders were excluded.

### Manual mask generation

We used LabKit 0.3.11, an ImageJ module extension, to manually generate the reference masks for our data sets. For pixel-level accuracy, we used pencil input (XP-PEN Artist13.3 Pro 13.3 Inch IPS drawing pen and monitor). Following the approach of reference (32), which confirmed alignment between cell membrane boundaries and image object boundaries, we produced masks that include the surrounding fading halo (i.e., shade-off) of each cell object.

### Manual tracking-trajectory generation

Ground truth tracking trajectories were established manually. The imaging frequency was sufficiently high that there were no ambiguities in assigning track links or parent links.

### Image processing

When using each package, one must choose which modules to incorporate and what values to assign to hyperparameters. The specific pipelines employed in this study are posted on Github and OSF.

### Image preprocessing

All raw images underwent preprocessing using the "unsharp mask" filter in ImageJ. Ilastik (1.4.0) preprocessing was applied to images supplied to CellProfiler's traditional segmentation pipeline and to FAST. The Ilastik training files are posted on Github and OSF. Ilastik pixel classification was applied to the large-population data sets. For the workable data sets, Ilastik object classification was also applied to produce output for subpixel image registration-based jitter correction, which was implemented through a Python script (Github) and OSF.

### Cell object counts

CellProfiler (and CP-Omnipose) provides object count directly. SuperSegger-Omnipose, DeLTA, and FAST provide output that must be processed to determine object counts. Custom scripts for this procedure are available at Github and OSF. Cell objects in contact with the borders were excluded from object counts. For CellProfiler and FAST,

this exclusion was handled by the software itself. For DeLTA and SuperSegger-Omnipose, custom-written scripts were produced for this purpose at Github and OSF.

When determining cell counts for the constrained growth case (Fig. S3), in some cases, the packages mistakenly assigned parts of the chamber walls as cell objects. These mistaken objects were manually excluded from the reported counts by alignment with coordinates that were confirmed as part of the chamber wall. (Alternatively, CellProfiler and FAST can process constrained environment images by removing microfluidic device walls in a preprocessing step with Ilastik; to maintain consistency, this approach was not used.)

### Intersection over union

Intersection over Union (IoU)-binary version is defined in terms of the overlap between the predicted segmentation (binary mask) and the ground truth reference object mask, considering only two pixel classes: object and background.

For each software tool, binary IoU was determined by comparison with the output mask. Scripts to generate this output from each tool, as well as a script to generate IoU, are accessible at Github and OSF along with all output reference mask data sets.

### Tracking performance and geometric features

The data for counting branches in cell lineages (Table 5), cell division cycle duration (Fig. 5), cell object length (Fig. 6), and elongation rate (Fig. 7) are not directly obtainable from the packages' outputs. Thus, post-processing scripts were developed to extract these data, including transformation of package output into accessible file formats, available at Github and OSF.

In assigning the number of branches in cell lineages, the ground truth was used to guide the assessment of package output. Each error reported in Table 5 is determined by comparing the ground truth lineage tree with an output graph constructed by starting with the ground truth lineage tree and removing all cell objects that were not assigned the correct lineage ID.

## RESULTS

Here, we present a comparison of the performance of the four packages under consideration. We applied segmentation, tracking, and feature assignment pipelines to time-lapse phase-contrast images of unconstrained monoculture growth of *E. coli* populations, both on coverslips under agar pads (unconstrained growth) and in a microfluidic trap (constrained growth). In addition, we assessed performance on time-lapse images of unconstrained populations of *P. putida* and *X. campestris*.

For each pipeline, we used developer-recommended parameter values and module choices (determined through direct recommendations from the package development teams). All images were pre-processed with the "unsharp mask" filter in ImageJ (42). In addition, images segmented by FAST and CellProfiler's traditional segmentation tools were pre-processed with Ilastik (43), a machine learning-based pixel segmentation tool, as recommended by the CellProfiler developers. Specifics of all pipelines applied (including Ilastik training files) are provided at Github and OSF.

While we have endeavored to provide a comparison that offers a generally useful perspective on package performance, we acknowledge that no particular assessment can provide a comprehensive characterization of a package's overall utility.

### Segmentation

#### Segmentation of densely populated images (over 1,000 objects)

##### Unconstrained growth environment

To assess segmentation performance, we used each package to process time-lapse images of an *E. coli* population that reaches high density and large population size (over

1,000 objects). This experimental time-lapse contains 100 timepoint images separated by 90-second intervals over a total of 150 minutes (details in Materials and Methods). Figure 2 shows the first and last time points of this data set. (The complete data set is posted on Github and OSF.) For this data set, we manually generated ground truth object counts for each frame (details in Materials and Methods).

Figure 3 shows the relative error and (inset) the number of objects identified at each timestep, along with the ground truth value. For all packages, as well as the ground truth, objects touching the borders were excluded. This exclusion was done manually for the ground truth and automatically within CellProfiler and FAST. For SuperSegger-Omnipose and DeLTA, scripts for this purpose are provided at Github and OSF.

All packages performed well in cell count segmentation, with DeLTA exhibiting the closest alignment to the ground truth counts, closely followed by FAST.

Object counts for complementary data sets (with final cell counts over 650) of unconstrained growth of *X. campestris* and *P. putida*, without reference ground truth, are provided in Fig. S1 (Appendix) and S2 (Appendix). (The data sets are posted on Github and OSF.) These show similar trends in terms of package performance.

Another metric for segmentation performance is the Intersection over Union measure (11), defined in terms of object masks. Construction of ground truth object masks can be subjective. Each cell object is surrounded by shade-off: a decision must be made as to how much of this boundary region should be included as part of the mask. A novel approach for establishing this boundary was pioneered in reference (32), in which strains engineered to fluoresce on their outer membrane were imaged. The resulting masks were confidently aligned with biological object boundaries. Despite the lack of such fluorescence markers in our images, we endeavored to assign masks using a corresponding boundary threshold. Consequently, the IoU results reported below are biased toward Omnipose, which assigns mask boundaries using the fluorescence-validated approach in reference (32).

IoU measures for the last time point of the unconstrained *E. coli* growth time-lapse (Fig. 2) are shown in Table 1, determined in reference to a manually generated ground truth reference mask (details in Materials and Methods). (A score of 1 indicates perfect agreement of object masks and 0 indicates no overlap.) All packages performed well, with Omnipose having an advantage in terms of reference mask production.

## Constrained growth environment

The constrained environments provided by microfluidic devices present specific challenges for image segmentation. Image artifacts, such as uneven illumination and halos, can appear in these contexts (44). Moreover, the boundaries of the growth chamber can be difficult to distinguish from cell objects. For all packages, we performed post-processing to manually exclude the boundary regions (details in Materials and Methods).

We assessed cell count segmentation performance on time-lapse images of an *E. coli* population in a microfluidic trap (details in Materials and Methods). This experimental time-lapse consists of 54 time point images separated by 180-second intervals, for a total of 162 minutes. Figure S3 (Appendix) shows the first and last time points of this data set. (The complete data set is posted on Github and OSF.) Figure S4 (Appendix) shows the number of objects detected in each timestep, as reported by each software package. The overall trend is similar to the unconstrained growth case (Fig. 3)

TABLE 1  Intersection over union measures of segmentation accuracy[a]

| CP-Omnipose | SS-Omnipose | DeLTA | FAST |
|---|---|---|---|
| 0.91 | 0.91 | 0.58 | 0.58 |

[a]CP, CellProfiler and SS, SuperSegger.

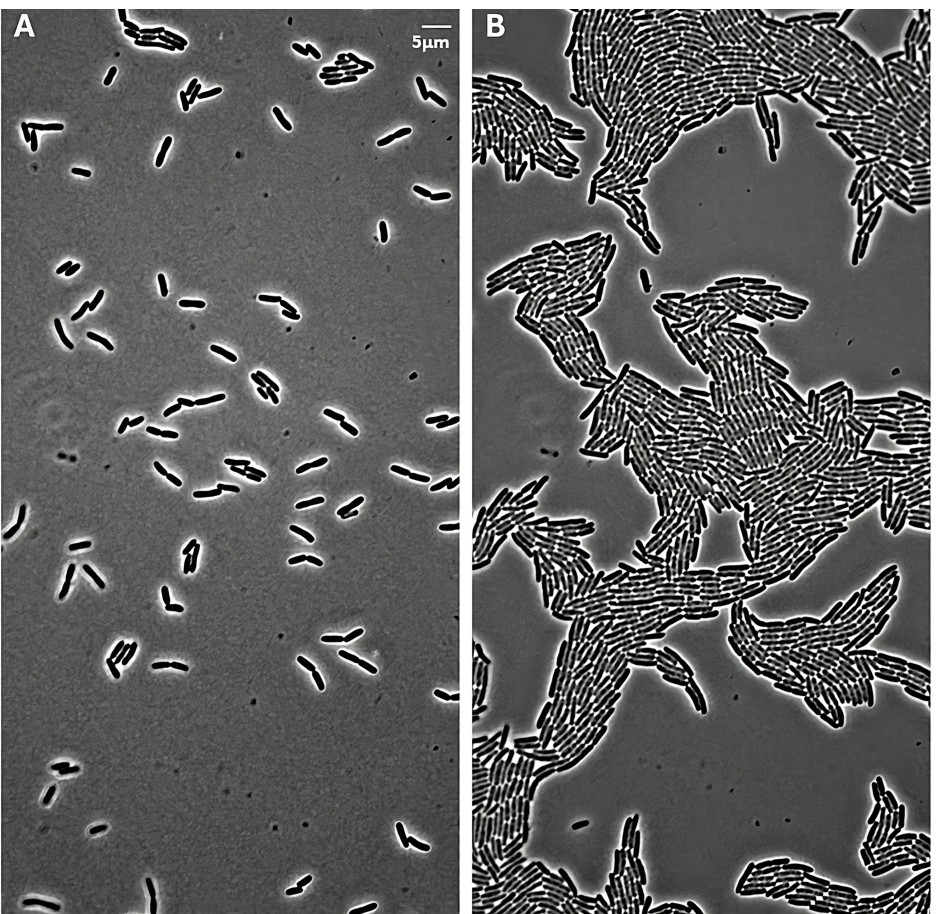

**FIG 2** Segmentation benchmarking data set. (A) First and (B) last (100th) images in a time-lapse of *E. coli* growth in an unconstrained environment. The initial frame contains 105 cells. The population grows to 1,140 by the final frame. Cells touching the edges were not counted. The cells are in monolayer throughout (150-minute duration).

## Workable time-lapses

We next consider the types of errors made in object identification. This analysis requires a manual comparison of the ground truth with the object identification results from each package. Application of such a manual analysis to the large data sets presented above is beyond the scope of this project. Instead, we generated 24 workable time-lapses of unconstrained *E. coli* population growth, each consisting of between 20 and 40 time points separated by 5-minute intervals (with final cell counts ranging from 6 to 47). We cropped these time-lapses to ensure that no cells intersect the boundary of any frame. Representative images from one of these workable *E. coli* time-lapses are shown in Fig. S5 (Appendix) (The complete time-lapses are posted on Github and OSF.).

We classify each segmentation error as either false positive or false negative. A false positive occurs when a non-cell pixel cluster (e.g., debris) is identified as a cell object or when a single cell object is mistakenly identified as two objects. A false negative occurs when a cell object is overlooked or when two neighboring cells are identified as one cell object.

Table 2 presents manually determined counts of segmentation error type totals across the last frame of each of the 24 workable time-lapses. The (manually determined) corresponding total object count is 583.

For completeness, we report average IoU measures for these 24 workable time-lapses across all time points in Table 3; these are consistent with the results in Table 1.

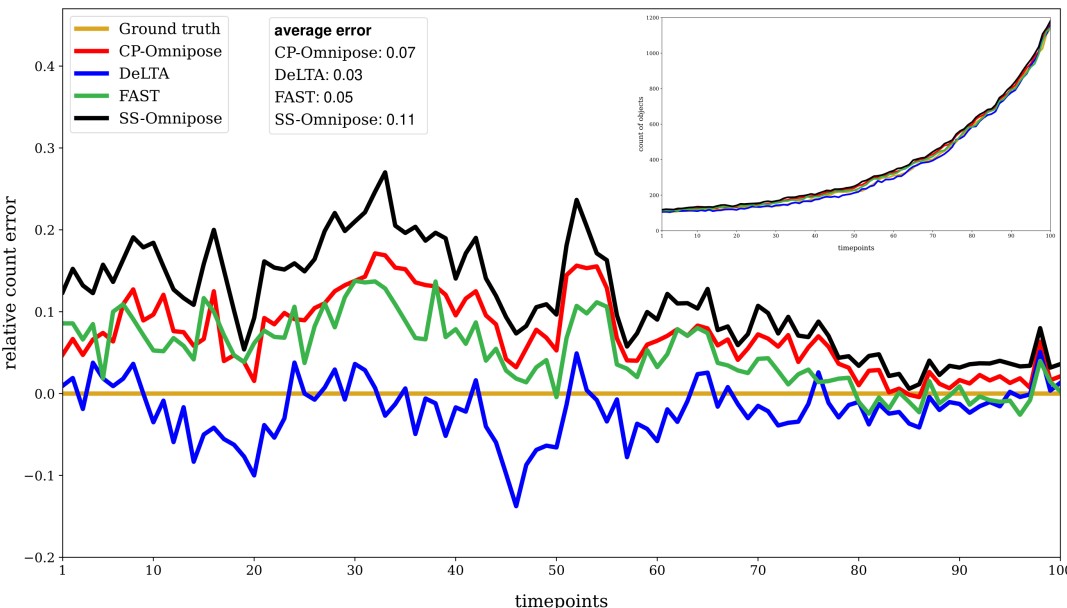

**FIG 3** Cell count segmentation results for time-lapse of unconstrained *E. coli* growth: relative error at each time point in comparison to the ground truth count. [Relative error = (estimate − ground truth)/(ground truth)]. The inset shows the object count for each tool along with the ground truth. Average error is obtained by averaging the absolute deviations from the ground truth across all timesteps. CP, CellProfiler and SS, SuperSegger.

## Tracking

Here, we present a comparison of tracking performance on the 24 workable time-lapses introduced above. For these data sets, no cells enter or leave the frame during the time-lapse.

### Tracking error counts

As discussed above, tracking involves identification of persistent objects (i.e., cells undergoing motion and growth) and mother–daughter triads (when division events occur). Each frame-to-frame connection is described as either a track link (persistence of object) or a parent link (mother-to-daughter). We follow Matula et al. (45) in establishing a classification scheme for tracking errors. Matula et al. provide a comprehensive error classification based on the identification of missing, redundant, or mislabeled (i.e., wrong semantic) links. Here, we introduce classifications only for the errors that we observed in the processing of the 24 workable data sets. Presuming no crossing of frame boundaries, tracking errors appear as one of the following (Fig. 4).

### Swapped track link

Mislabeling of object identities between two consecutive timesteps by swapping labels among a group of cells (typically two). These errors produce cross-overs in the forest of tracking trees.

### Missing track link

Initiating a spurious lineage by misidentifying an existing cell as a new immigrant. These errors introduce false lineage roots.

### Redundant parent link

Inappropriately assigning a cell as a daughter in a lineage where a division event did not occur. These errors create false divisions in the tracking tree.

**TABLE 2** Object identification error counts[a]

| Error type | CP-O | SS-O | DeLTA | FAST |
|---|---|---|---|---|
| False positive | 22 | 25 | 14 | 6 |
| False negative | 19 | 18 | 39 | 30 |

[a]CP, CellProfiler; SS, SuperSegger; and O, Omnipose.

### Missing parent link

Initiating a spurious lineage by failing to connect a new daughter cell in lineage with its mother. These errors introduce false lineage roots.

Table 4 shows the tracking error counts for the last timestep of the 24 workable time-lapses, as assessed against the manually determined ground truth. A total of 583 objects were tracked.

### Division detection

Segmentation and tracking errors contribute to inaccuracies in constructed cell lineages. We assessed the accuracy of lineage construction by comparing the number of division events (i.e., branch points) in lineage trees generated from each ancestor cell present in the first time frame of each timelapse. We manually generated ground truth lineages for each such ancestor. Table 5 shows a comparison of the average relative error in the number of branch points reported by each package (over the 38 lineages in the 24 workable data sets, details in Materials and Methods).

### Cell division cycle duration

Errors in tracking and segmentation lead to inaccurate characterizations of cell division cycle duration (i.e., the time elapsed from a cell's birth to division). Figure 5 shows the distribution of cell division cycle duration for the 24 workable *E. coli* data sets, as generated by the four packages, along with the manually determined ground truth (details in Materials and Methods). Only complete cell division cycles (observed from birth to division) are included.

## Geometric feature assignment

As discussed above, the image processing packages employ a range of strategies to assign object features, such as position, length, and elongation rate. Developing a manually determined ground truth for such features is beyond the scope of this project. We present below a direct comparison of feature assignment results for the four packages.

### Cell object length

Figure 6 shows the distribution of cell lengths collected across all time points of the 24 workable *E. coli* time-lapse data sets.

### Elongation rate

Figure 7 shows the distribution of elongation rates collected across all time points of the 24 workable *E. coli* data sets.

**TABLE 3** Average IoU over the 24 workable *E. coli* time-lapses[a]

| CP-Omnipose | SS-Omnipose | DeLTA | FAST |
|---|---|---|---|
| 0.82 | 0.85 | 0.66 | 0.69 |

[a]CP, CellProfiler and SS, SuperSegger.

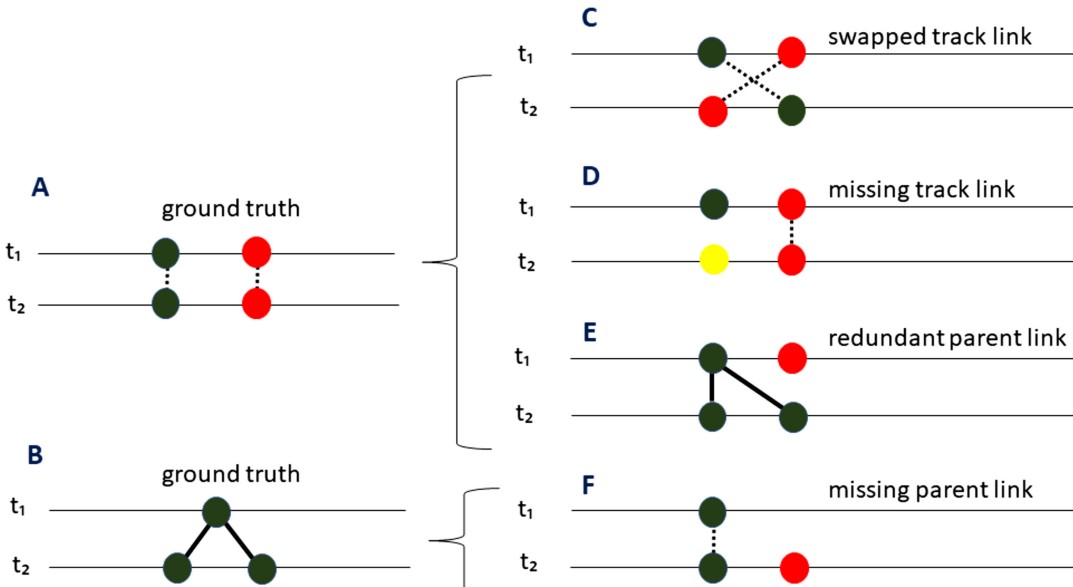

**FIG 4** Classification of tracking errors. Ground truth consists of (A) two persistent cells; (B) a mother–daughter triad. Errors: (C) swapped track link; (D) missing track link; (E) redundant parent link; and (F) missing parent link. Dotted edges: track links; solid edges: parent links. Colors represent distinct (identified) lineages.

## Execution time

Table 6 shows the execution time for each imaging pipeline (including segmentation, tracking, and feature assignment; details posted on Github and OSF) on time-lapse data sets of *E. coli* population growth (i) in the unconstrained environment (Fig. 2), (ii) the constrained environment (Fig. S3 Appendix), and (iii) in the 24 workable time-lapses. These do not include time spent manually training Ilastik (used for FAST), which can be significant. This analysis was performed exclusively using a CPU (details in Materials and Methods), without taking advantage of the GPU acceleration that is available for DeLTA.

## Robustness with respect to image noise

The results presented above are based on recommended practice for optimal performance. Image quality is impacted by experimental procedures and microscopy settings, which, in some cases, may lead to sub-optimal resolution. For completeness, we assessed cell count segmentation performance for an unfiltered version of the unconstrained *E. coli* population growth data set shown in Fig. 2, which contains artifacts introduced by inhomogeneities in the environment. [Corresponding first and last time point images are shown in Fig. S6 (Appendix); the full data set is available at Github and OSF.] Because deep learning segmentation algorithms can be sensitive to noise types on which they have had limited training (46), we included CellProfiler with traditional segmentation (and Ilastik input) in this analysis. The results are shown in Fig. 8.

**TABLE 4** Tracking errors counts[a]

| Tracking error type | CP-O | SS-O | DeLTA | FAST |
|---|---|---|---|---|
| Swapped track link | 5 | 8 | 7 | 10 |
| Missing track link | 6 | 8 | 6 | 11 |
| Missing parent link | 9 | 9 | 6 | 10 |
| Redundant parent link | 5 | 3 | 1 | 8 |

[a]CP, CellProfiler; SS, SuperSegger; and O, Omnipose.

**TABLE 5** Average relative error in division detection

| CP-Omnipose | SS-Omnipose | DeLTA | FAST |
|---|---|---|---|
| 29.66 | 20.12 | 32.88 | 53.21 |

*a*CP, CellProfiler and SS, SuperSegger.

## DISCUSSION

The analysis presented here involved the assessment of 1,094 images, representing 118,722 individual cells of three different bacterial species (Full data sets at Github and OSF.).

Considering cell count segmentation performance (Fig. 3), DeLTA and FAST showed the closest alignment with the ground truth. All packages exhibited improved performance at later time points. This could be attributed to the fact that, when fewer cells are in the frame, there is more opportunity for background noise to be identified as cell objects by the deep learning-based algorithms. Table 2 indicates that the deep learning segmentation algorithms (Omnipose and DeLTA) produce the most false positives, with DeLTA performing better than Omnipose. In comparing (unbenchmarked) cell count segmentation performance of *E. coli* population growth in a constrained environment and unconstrained growth of *P. putida* and *X. campestris* populations, similar trends appeared.

When assessing the robustness of the results with respect to image quality, we found that traditional segmentation approaches coupled with Ilastik yielded superior results for cell count segmentation performance. This is consistent with reports that deep learning segmentation algorithms suffer from noise sensitivity (46) depending on the choice of training set. In the related field of medical imaging, significant efforts have been made toward algorithm refinement to address this limitation (47). One such solution is the incorporation of a Gaussian kernel. This option is provided by DeLTA and Omnipose, although the results on unfiltered images (Fig. 8) suggest that further refinements may yet improve performance.

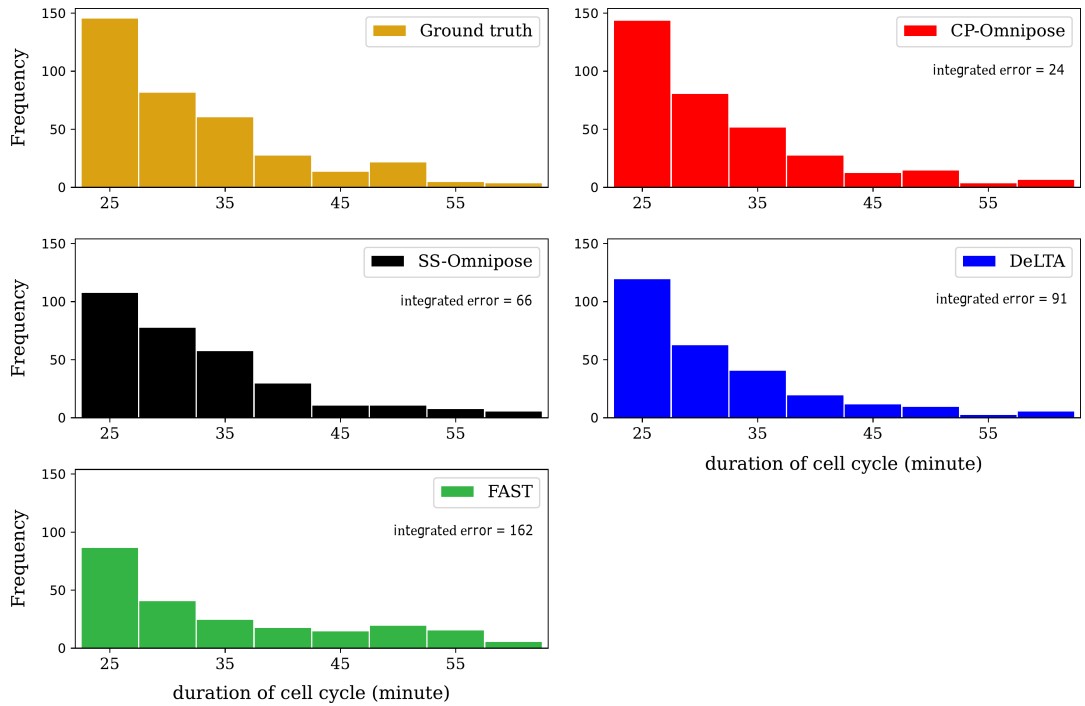

**FIG 5** Distribution of cell division cycle duration for the 24 workable *E. coli* data sets. Integrated error describes the accumulated absolute error between each package's output and the ground truth. (Incomplete cell cycles are not included.) CP, CellProfiler and SS, SuperSegger.

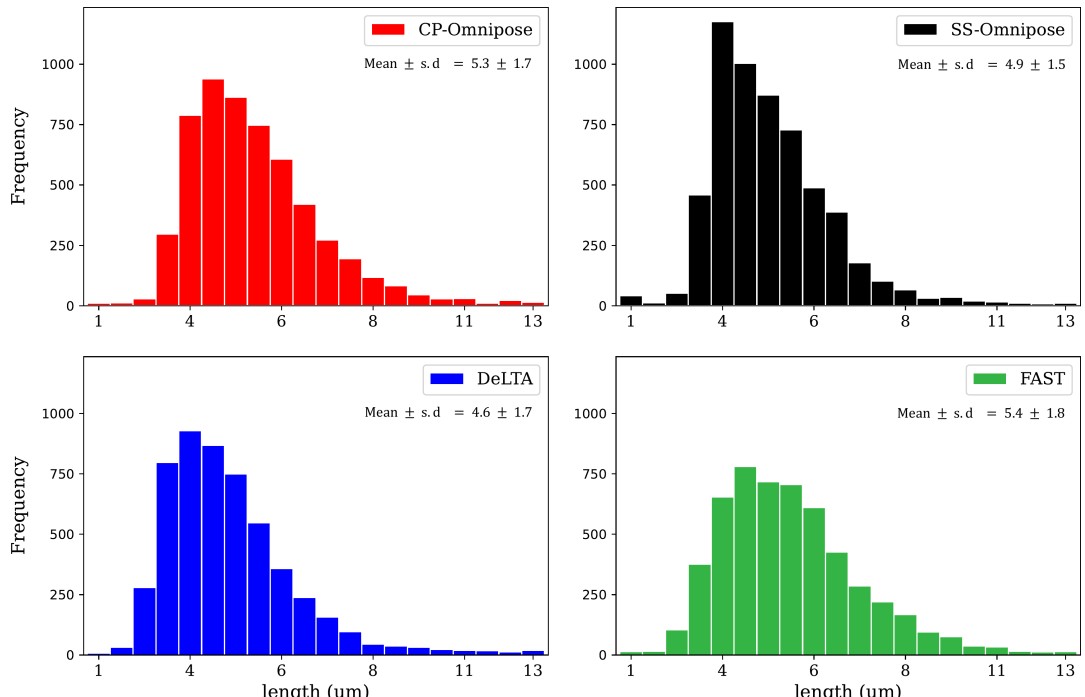

**FIG 6** Distribution of cell object lengths gathered from all time points across the 24 workable *E. coli* data sets. CP, CellProfiler and SS, SuperSegger.

Comparison of IoU metrics presents a challenge due to the subjective nature of manual annotation, particularly when defining object boundaries. The ambiguity arises from the presence of a "halo" around each object, leading to uncertainty about whether to include this area as part of the object.

Our analysis of tracking errors (Table 4) reveals that package performance is reasonably consistent. The most commonly occurring errors are missing track links and missing parent links. In reconstructing cell tracking trajectories and associated lineages (Table 5), we found that SuperSegger-Omnipose most closely aligns with the ground truth, followed by CP-Omnipose and DeLTA. In the corresponding assessment of cell cycle duration, CP-Omnipose matches the ground truth well, followed by Supersegger-Omnipose and DeLTA. Finally, in comparing (unbenchmarked) geometric feature assignment, the packages show reasonable agreement.

In terms of execution times, the packages demonstrated similar performance on the smaller, workable data sets. However, for the much larger benchmarked *E. coli* data set, FAST demonstrated much faster execution than the rest. We did not assess potential gains in GPU-based parallelization.

The analysis presented above required custom scripts to generate consistent outputs from all packages. These include the exclusion of objects touching the perimeters (including consideration of mask padding by deep learning algorithms), generation of output masks, and transformation of package output into a structured .csv file containing cell attributes. In addition, we developed custom scripts to support generation of the ground truth annotations of object counts, masks, and tracking trajectories, and for the analysis of life-history-based measures. These scripts could be used to empower future analyses or pipeline development; likewise, the ground truth annotations may be useful

**TABLE 6** Intersection over union measures of segmentation accuracy[a]

| CP-Omnipose | SS-Omnipose | DeLTA | FAST |
|---|---|---|---|
| 0.91 | 0.91 | 0.58 | 0.58 |

[a]CP, CellProfiler and SS, SuperSegger.

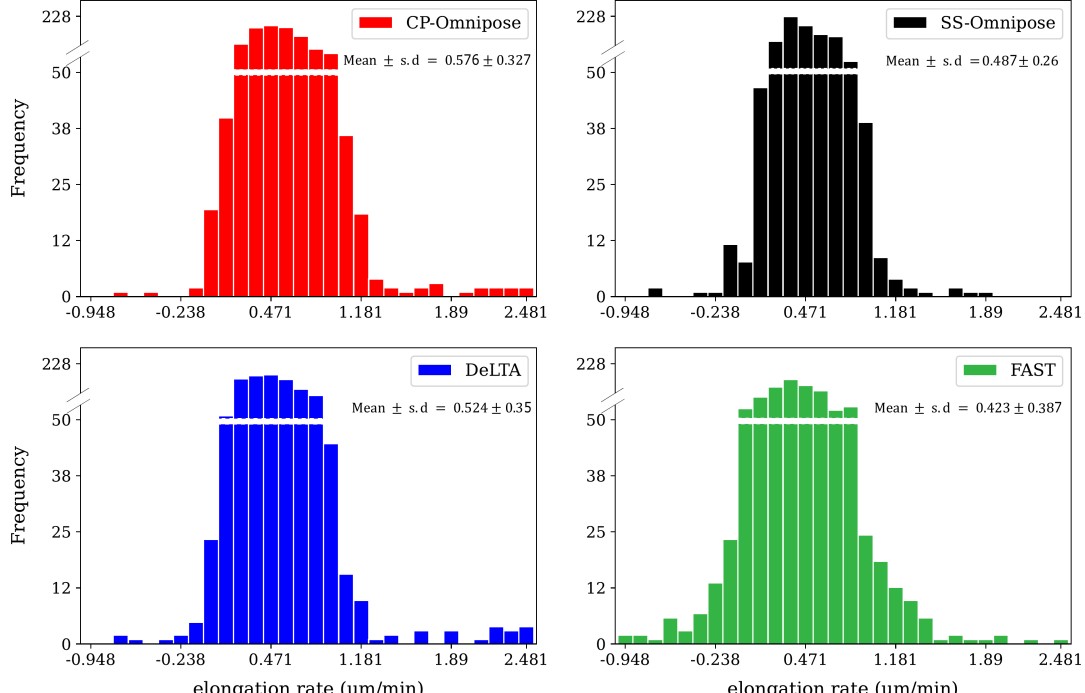

**FIG 7** Distribution of elongation rates collected from all time points of the 24 workable *E. coli* data sets. Elongation rate is defined as $(L_t + 1 - L_t)/\Delta t$, where $L_t$ is length (in µm) of the cell object in frame $t$ and $\Delta t = 5$ minutes is the interval length. CP CellProfiler and SS, SuperSegger.

for benchmarking future performance comparisons. All of this material is available at Github and OSF.

We acknowledge that our performance comparison does not capture the full range of package capabilities. For instance, our focus on rod-shaped bacteria neglected consideration of the rich diversity of cell morphologies; likewise, we did not consider populations of cells undergoing self-propelled motility. In terms of data analysis, we did not compare across imaging or pre-processing techniques [such as deconvolution (48) or jitter correction options] nor did we take advantage of the improved segmentation accuracy that would be provided by fluorescent labeling of cell membranes (32). Moreover, we did not provide a comprehensive comparison across pre- or post-processing software add-ons, nor did our analysis capture the full spectrum of metrics that could be of interest for a given imaging project.

The four packages we considered share common features, such as customizability, cross-platform compatibility, and the ability to handle large-scale data sets. Moreover, each package offers unique features that may be essential for particular research tasks. CellProfiler can measure confluence in cell culture experiments and identify subcellular compartments (such as nuclei, cytoplasm, and organelles). Moreover, the machine learning-based module CellProfiler Analyst allows for interactive exploration and classification of image data (31). SuperSegger-Omnipose can directly assess the spatial distribution of bacterial cells in an image, identifying patterns of cell arrangement, clustering, or organization, including the structure and growth of bacterial colonies. DeLTA's deep learning model can be trained on a wide range of data sets, making it adaptable to a range of bacterial species, growth conditions, and imaging modalities, and offering the potential to detect and analyze subcellular structures. FAST self-reports a "trackability" score—an assessment of the reliability and accuracy of tracking, based on object features, motion characteristics, and object similarity between consecutive frames.

As discussed in references (11, 49), deep learning-based packages have transformed the field of image processing (50). These tools can efficiently handle larger data sets

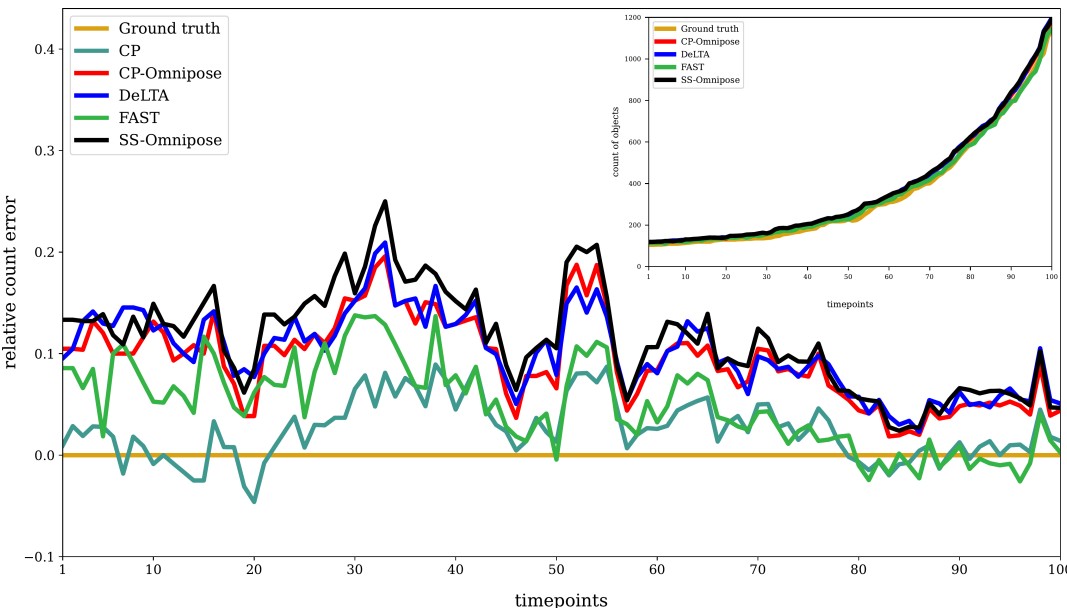

**FIG 8** Segmentation cell count for unfiltered time-lapse of unconstrained *E. coli* growth. Relative error with respect to ground truth. The inset shows the cell object counts.

[in comparison to traditional methods (51)] and can be adapted to work well on a wide range of tasks and data types, providing versatile, all-purpose solutions (52, 53). A key advantage of deep learning algorithms is their ability to automatically learn useful features from data, which reduces the need for manual feature engineering and expert input (54). The results reported here provide clear evidence of the value of deep learning tools in processing images of monolayer bacilli, especially for object segmentation. However, our analysis shows that traditional approaches to tracking (currently) offer superior performance. Combination pipelines thus appear to offer the best approach; future developments in deep learning strategies may tip the balance in their favor.

## ACKNOWLEDGMENTS

We would like to express our gratitude to Beth Cimini, Rebecca Senft, Paul A. Wiggins, Kevin Cutler, Teresa Lo, Daniela Koch, Mary Dunlop, Owen O'Connor, Virgile Andreani, Jean-Baptiste Lugagne, William M. Durham, Oliver J. Meacock, Christine Jacobs-Wagner, Sander Govers, and Yingjie Xian for their valuable contributions and insights in this study.

This work was supported by a Discovery Grant (RGPIN-2018-03826) from Canada's Natural Sciences and Engineering Research Council (NSERC). This work was carried out on the Haldimand Tract, land granted to the Haudenosaunee (Six Nations) in 1784. Settler colonial theft of this land is ongoing.

## AUTHOR AFFILIATIONS

[1]Department of Biology, University of Waterloo, Waterloo, Ontario, Canada
[2]Department of Mechanical and Mechatronics Engineering, University of Waterloo, Waterloo, Ontario, Canada
[3]Department of Applied Mathematics, University of Waterloo, Waterloo, Ontario, Canada

## AUTHOR ORCIDs

Atiyeh Ahmadi  http://orcid.org/0000-0002-2303-6730
Brian Ingalls  http://orcid.org/0000-0003-2118-3881

## FUNDING

| Funder | Grant(s) | Author(s) |
|---|---|---|
| Canadian Government \| Natural Sciences and Engineering Research Council of Canada (NSERC) | RGPIN-03826-2018 | Brian Ingalls |
| Canadian Government \| Natural Sciences and Engineering Research Council of Canada (NSERC) | RGPIN-04151-2018 | Carolyn Ren |

## AUTHOR CONTRIBUTIONS

Atiyeh Ahmadi, Conceptualization, Formal analysis, Investigation, Methodology, Software, Validation, Visualization, Writing – original draft, Writing – review and editing | Matthew Courtney, Methodology, Resources | Carolyn Ren, Resources | Brian Ingalls, Conceptualization, Funding acquisition, Supervision, Writing – review and editing

## DATA AVAILABILITY

The data sets produced and the scripts developed for conducting the analyses are available at Github and OSF.

## ADDITIONAL FILES

The following material is available online.

### Supplemental Material

**Supplemental material (Spectrum00032-24-s0001.pdf).** Supplemental table and figures.

### Open Peer Review

**PEER REVIEW HISTORY (review-history.pdf).** An accounting of the reviewer comments and feedback.

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
