## [Reviewer comments · Microbiology Spectrum]

Microbiology Spectrum

A benchmarked comparison of software packages for time-lapse image processing of monolayer bacterial population dynamics

Atiyeh Ahmadi, Matt Courtney, Carolyn Ren, and Brian Ingalls

Corresponding Author(s): Brian Ingalls, University of Waterloo

Review Timeline:

Submission Date:

January 4, 2024

Accepted:

April 26, 2024

Editor: Erik Hom

Reviewer(s): Disclosure of reviewer identity is with reference to reviewer comments included in decision letter(s). The following individuals involved in review of your submission have agreed to reveal their identity: Kevin J. Cutler (Reviewer #1)

Transaction Report:

DOI: <https://doi.org/10.1128/spectrum.00032-24>

Re: Spectrum00032-24 (A benchmarked comparison of software packages for time-lapse image processing of monolayer bacterial population dynamics)

Dear Prof. Brian Ingalls:

Thank you for your patience. I have gone ahead and made a decision to accept your manuscript based on the reviews I received. There is some suggestions to try to improve your figures and perhaps some of your text (and see Reviewer #1 comments) but I feel comfortable accepting your manuscript. If you could improve upon what you have in the proofing stage, I encourage you to do so.

I am forwarding your manuscript to the ASM production staff for publication. Your paper will first be checked to make sure all elements meet the technical requirements. ASM staff will contact you if anything needs to be revised before copyediting and production can begin. Otherwise, you will be notified when your proofs are ready to be viewed.

Sincerely,
Erik Hom
Editor
Microbiology Spectrum

Reviewer #1 (Public repository details (Required)):

The data is already available on GitHub. Perhaps an OSF copy would also be appropriate.

Reviewer #1 (Comments for the Author):

This manuscript and associated code provide a much-needed quantitative comparison among leading microbe segmentation and tracking approaches. The curation of a ground-truth dataset and complete, well-documented benchmarking code is a truly impressive accomplishment. This will be informative to any researchers getting started on quantitative microscopy with bacteria under phase contrast and should generalize to other organisms and imaging modalities.

Comments on data:

As stated, any amount of benchmarking data is valuable to the community. However, there is room for improvement in the provided datasets.

(1) Time lapses should always be corrected for jitter and drift. As-is, the workable E. coli dataset is difficult to manually validate and the jitter could throw off tracking algorithms.

(2) There are some very high-quality datasets in the associated GitHub repository, but they were not used as the basis for ground-truth data. Those images that use only a few of the 16 (or 8) bits available during acquisition are more difficult to interpret against their ground-truth labels.

No fluorescence channels were used to validate cell segmentation in phase. In particular, no marker for cell division was used, and so human annotations might be too early, too late, or have much high variability about the true division time. This may have an impact on tracking results and division time measurements. Likewise, the lack of fluorescent membrane or cytosol labels makes the ground truth annotations much more subjective than they need to be.

Overall, I have confidence that the data and annotations are of sufficient quality to support the conclusions of this particular study. However, some discussion of the potential weaknesses and areas of improvement would be valuable.

Comments on figures:

Use semantic gamma normalization. All micrographs should look closer to Fig S6.

Visual comparisons of segmentation between packages would be useful.

Fig 1: crop out of empty space. Replace low-res 3 μ m marker. Fix offset arrows.

Fig 2: Image normalization seems off, hard to see detail inside and between cells. Cells are too dark and halo is too bright.

Figs 6, 7: a statistical measure of similarity between distributions could be useful.

Fig S4: use log₂ scale for cell count.

Reviewer #3 (Comments for the Author):

After reviewing the paper, the overall quality of your work is commendable. The clarity and coherence of the introduction were particularly impressive as it helps understand this specific field of research. You provided a comprehensive overview of CellProfiler, FAST, DeLTA, and SuperSegger-Omnipose, making it accessible to readers unfamiliar with software algorithms.

Your explanation of segmentation, tracking, and feature assignment was clear and served as a strong foundation for understanding the subsequent sections of the paper. This approach greatly enhances the accessibility of your research to a broader audience within the scientific community.

Furthermore, the logical structure of the results section was effective. The presentation of data followed a coherent pattern, allowing readers to easily follow the analysis and conclusions drawn from comparing the four software packages. The significance of the findings in aiding scientists to make informed decisions regarding the selection of appropriate models was clearly demonstrated.

Overall, your paper makes a valuable contribution to the field and will be of great interest to researchers working in this area. Some minor suggestions for improvement have been provided in the detailed review, but it is important to note that these do not detract from the overall excellence of your work.

Thank you for the opportunity to review your paper, and I look forward to seeing it published.